# Timing of exposure is critical in a highly sensitive model of SARS-CoV-2 transmission

Ketaki Ganti[1], Lucas M. Ferreri[1], Chung-Young Lee[1], Camden R. Bair[1], Gabrielle K. Delima[1‡], Kate E. Holmes[1‡], Mehul S. Suthar[1,2,3,4], Anice C. Lowen[1,4]*

1 Department of Microbiology and Immunology, Emory University School of Medicine, Atlanta, Georgia, United States of America, 2 Emory Vaccine Center, Emory University School of Medicine, Atlanta, Georgia, United States of America, 3 Center for Childhood Infections and Vaccines of Children's Healthcare of Atlanta, Department of Pediatrics, Emory University School of Medicine, Atlanta, Georgia, United States of America, 4 Emory-UGA Center of Excellence for Influenza Research and Surveillance [CEIRS], Atlanta, Georgia, United States of America

☺ These authors contributed equally to this work.
‡ These individuals contributed equally to the research presented.
* anice.lowen@emory.edu

**Data Availability Statement:** All relevant data are within the manuscript and its Supporting Information files.

## Abstract

Transmission efficiency is a critical factor determining the size of an outbreak of infectious disease. Indeed, the propensity of SARS-CoV-2 to transmit among humans precipitated and continues to sustain the COVID-19 pandemic. Nevertheless, the number of new cases among contacts is highly variable and underlying reasons for wide-ranging transmission outcomes remain unclear. Here, we evaluated viral spread in golden Syrian hamsters to define the impact of temporal and environmental conditions on the efficiency of SARS-CoV-2 transmission through the air. Our data show that exposure periods as brief as one hour are sufficient to support robust transmission. However, the timing after infection is critical for transmission success, with the highest frequency of transmission to contacts occurring at times of peak viral load in the donor animals. Relative humidity and temperature had no detectable impact on transmission when exposures were carried out with optimal timing and high inoculation dose. However, contrary to expectation, trends observed with sub-optimal exposure timing and lower inoculation dose suggest improved transmission at high relative humidity or high temperature. In sum, among the conditions tested, our data reveal the timing of exposure to be the strongest determinant of SARS-CoV-2 transmission success and implicate viral load as an important driver of transmission.

## Author summary

Interrupting SARS-CoV-2 transmission is a major goal of non-pharmaceutical interventions designed to mitigate the impact of the COVID-19 pandemic. To support these efforts, fundamental research on SARS-CoV-2 transmission is greatly needed. With this goal in mind, we used a golden Syrian hamster model to evaluate the extent to which the timing of exposure, ambient temperature and ambient humidity modulate the efficiency of SARS-CoV-2 transmission through the air. Our data indicate that robust transmission

**Funding:** This work was funded by the National Institute of Allergy and Infectious Disease through the Centers of Excellence for Influenza Research and Surveillance (CEIRS) contract no. HHSN272201400004C (ACL). GKD is supported by F31 AI 50114. The funders had no role in study design, data collection and analysis, decision to publish, or preparation of the manuscript.

**Competing interests:** The authors have declared that no competing interests exist.

is maintained with exposure durations as short as one hour and across a wide range of humidity and temperature conditions. Conversely, the timing of exposure relative to infection of donor animals was found to strongly impact the likelihood of transmission, with exposure prior to 16 hours and after 48 hours post-inoculation yielding little or no spread to contacts. Importantly, the likelihood of successful transmission corresponded with infectious viral titers in the nasal tract, strongly suggesting that the infectious period is defined by the dynamics of viral load in donor hosts.

## Introduction

The COVID-19 pandemic has led to a public health emergency and social disruption on a scale not seen since the influenza pandemic of 1918. Non-pharmaceutical interventions have been pursued in many parts of the world with the goal of limiting the impact of the outbreak [1–3]. These interventions seek to interrupt transmission of the virus [1–3]. Effective use of non-pharmaceutical interventions therefore relies heavily on fundamental understanding of SARS-CoV-2 transmission and the viral, host, and environmental factors that modulate its efficiency.

For example, efforts to contain viral spread through contract tracing rely on estimates of the duration of exposure needed to support transmission. In defining a close contact, exposure to an infected individual for a minimum of 15 or 30 minutes is often used [4,5]. In turn, quarantine measures for those identified as a close contacts rely on estimates of incubation period, commonly reported as 10–14 days [6,7]. Similarly, policies for isolation of positive cases reflect the potential for onward transmission within a period up to 10- or 14-days post-infection [6,7]. Evidence-based refinement of these definitions is important for establishing infection-control practices that minimize risk of transmission while also mitigating the social and economic burden of quarantine measures.

Much attention has also been paid to the potential for various environmental conditions to modulate the efficiency of transmission. Epidemiological investigation of environmental parameters has returned varied results [8–10]. Low temperature was found to be associated with increased SARS-CoV-2 transmission in some cases [10,11] but not others [12,13]. Similarly, dry conditions were found to be favorable for SARS-CoV-2 spread in a subset of studies [11,14]. In general, detected effects of temperature and humidity on reproduction number or epidemic growth were dwarfed by those of active interventions such as restrictions on mass gatherings [13,15,16]. Indeed, the complexity of conditions under which human exposures occur, and incomplete information regarding those conditions, can frustrate efforts to define parameters important for transmission efficiency. Examination of transmission in relevant experimental models therefore forms an invaluable complement to epidemiology.

Golden Syrian hamsters are highly susceptible to infection with SARS-CoV-2, show clear signs of disease, shed the virus at high titers from the upper respiratory tract, and transmit SARS-CoV-2 to direct contacts and through the air [17–19]. This model species has been used to evaluate the utility of vaccines, drugs, and passively transferred antibodies for blocking transmission [19–21]. Nevertheless, an important limitation of the model is that its extreme permissiveness can interfere with the detection of differences in transmission efficiency. To date, analysis of the fitness of novel variants has relied on their co-infection with ancestral strains [22–24]. While sensitive, this approach introduces the likelihood of viral fitness being modulated by interactions between the co-infecting viruses [25,26]. There is therefore a need for refinement of the hamster model to increase the stringency of transmission. Identification

of temporal and environmental factors that modulate transmission can help to achieve this goal.

Here, we used a hamster model to investigate the impact of a range of temporal and environmental conditions on SARS-CoV-2 transmission. Our data indicate that high transmission efficiency is maintained with exposure durations as short as one hour and across a wide range of humidity and temperature conditions. Conversely, the timing of exposure relative to infection of donor animals was found to strongly modulate transmission, with exposure prior to 16 h and after 48 hours post-inoculation [hpi] yielding little or no spread to contacts. The temporal structure of transmission corresponded with infectious viral titers in the nasal tract, strongly suggesting that the infectious period is defined by the dynamics of viral load in donor hosts.

## Results

### Minimal impact of exposure duration on SARS-CoV-2 transmission

Reasoning that viral densities in the air or the recipient respiratory tract may accumulate over time, we hypothesized that the period during which naïve animals are exposed to the exhaled breath of infected animals would impact the efficiency of transmission. To test this hypothesis, we used an exposure system in which naïve hamsters are placed on the opposite side of a porous, double-walled, barrier from infected hamsters for a defined duration [S1 Fig]. Exposures were carried out under controlled environmental conditions at 24 hpi of donor animals for periods of 120 h, 8 h, 4 h and 1 h. After exposure, animals were singly housed. Collection of nasal lavage samples was used to monitor for transmission and daily measurement of body weight was used to monitor clinical signs. A schematic of the transmission experimental setup is shown in Fig 1A. The results revealed robust transmission for all exposure durations tested [Fig 1B]. Positivity in a subset of contact animals was detected at the first sampling time of 2 d post-inoculation [1 d post-exposure] and all contact animals were found to harbor infectious

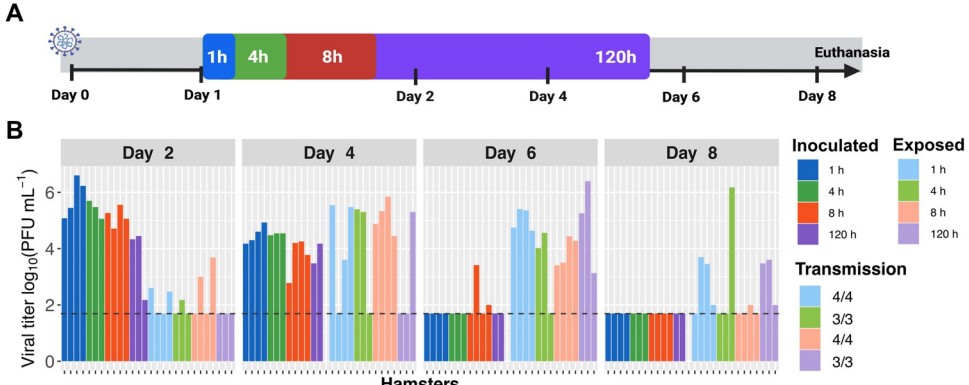

**Fig 1. Exposure durations as short as 1 h are sufficient for robust SARS-CoV-2 transmission.** A] Schematic depicting experimental timeline. Animals were inoculated with 1x10^4 PFU [titered on VeroE6 cells] at Day 0 and exposures initiated at Day 1. Colored boxes show the different exposure periods, with the duration indicated within each box. B] Viral titers in nasal lavage samples collected from inoculated [dark colors] and exposed [light colors] hamsters are plotted. Facets show results from 2-, 4-, 6- and 8-days post-inoculation [dpi] with different colors indicating the duration of exposure [1, 4, 8 or 120 h]. Exposures were carried out at 30˚C and 50% RH. Horizontal dashed line indicates limit of detection [50 PFU]. Missing data indicate that the animal died or was euthanized midway through the experiment. The fraction of transmission pairs in which recipients shed infectious virus at one or more time points is indicated at the right.

virus by 8 d post-inoculation [7 d post-exposure]. The period of exposure did not give rise to clear temporal trends in transmission. Thus, within the range tested, the period of exposure had minimal impact on SARS-CoV-2 transmission in hamsters. The data indicate that a minimal infectious dose is readily transferred to recipient hamsters within the course of one hour.

In inoculated animals, body weight loss was typically moderate to severe. While results were more variable for contact animals, the timing of their weight loss usually corresponded to the timing with which infectious virus was first detected in nasal washes [**S2 Fig**]. Occasionally, infected hamsters lost greater than 25% of their initial body weight and were euthanized.

## Timing of exposure strongly modulates SARS-CoV-2 transmission

We next tested the impact of the timing of exposure on SARS-CoV-2 transmission. To test for transmission at early times, brief exposures of one or two hours were carried out at 10–12, 12–14, 14–16, 16–17 or 24–25 hpi. Similarly, to evaluate transmission potential late in the course of infection, exposures were carried out for two hours on days 2, 4 and 6 post-inoculation. Nasal lavage samples collected from donor animals at the conclusion of each exposure window were used to assess viral loads at the time of exposure, while serial samples collected from these same donors across multiple time points were used to evaluate the dynamics of viral load.

A schematic of the transmission experimental setup is shown in **Fig 2A**. Analysis of nasal viral load in donor animals sampled at the time of exposure revealed a sharp increase between 12 h and 25 hpi, from an average of less than $1x10^3$ PFU/ml at 12 h to approximately $1x10^7$ PFU/ml at 25 h. Loads then declined over 2-, 4- and 6-days post-inoculation [dpi], averaging about $1x10^6$ PFU/ml on day 2 and declining to $< 1x10^2$ PFU/ml by day 6 [**Fig 2B**]. Dynamics of viral load across all inoculated animals sampled on days 2, 4, 6 and 8 were consistent with these results obtained at the conclusion of specific exposures and revealed that titers were typically below the limit of detection by 8 dpi [**S3 Fig**]. Taken together, the results show high viral loads of $1x10^5 – 1x10^7$ PFU/ml are reached by 17 hpi and sustained at 48 hpi, but that titers fall below this range by 4 dpi.

Analysis of exposed animals revealed a strong impact of the timing of exposure on transmission [**Fig 2C**]. Naïve animals exposed for 2 h ending at 12, 14 or 16 hpi of donors were not infected. By contrast, transmission was seen in two of four animals exposed for 1 h ending at 17 hpi and all four animals exposed for 1 h ending at 25 hpi. Exposure at late time points showed appreciable transmission only on day 2 post-inoculation, with five of eight hamsters contracting infection when exposed for 2 h beginning at 2 dpi. Overall, the data reveal a window of transmissibility from 17 h to 2 d after infection of the donor hamster [**Fig 2D**].

Again, in this experiment, body weight loss was typically moderate to severe in inoculated animals. While results were more variable for contact animals, for those hamsters that contracted infection, the timing of weight loss usually corresponded to the timing with which infectious virus was detected in nasal washes [**S4 Fig**].

To test whether viral load is a likely determinant of infectious period, nasal lavage titers in donor animals at the conclusion of exposure were evaluated. Significantly higher viral loads were detected in hamsters that transmitted to contacts, compared to those that did not transmit SARS-CoV-2 [**Fig 3**]. The data are consistent with a threshold viral titer of approximately $1x10^5$ PFU/ml needed for transmission.

## Minimal impact of humidity on SARS-CoV-2 transmission

Ambient humidity is known to modulate the efficiency of aerosol transmission of influenza A virus, with dry conditions favoring spread [27–29]. We hypothesized that a similar effect

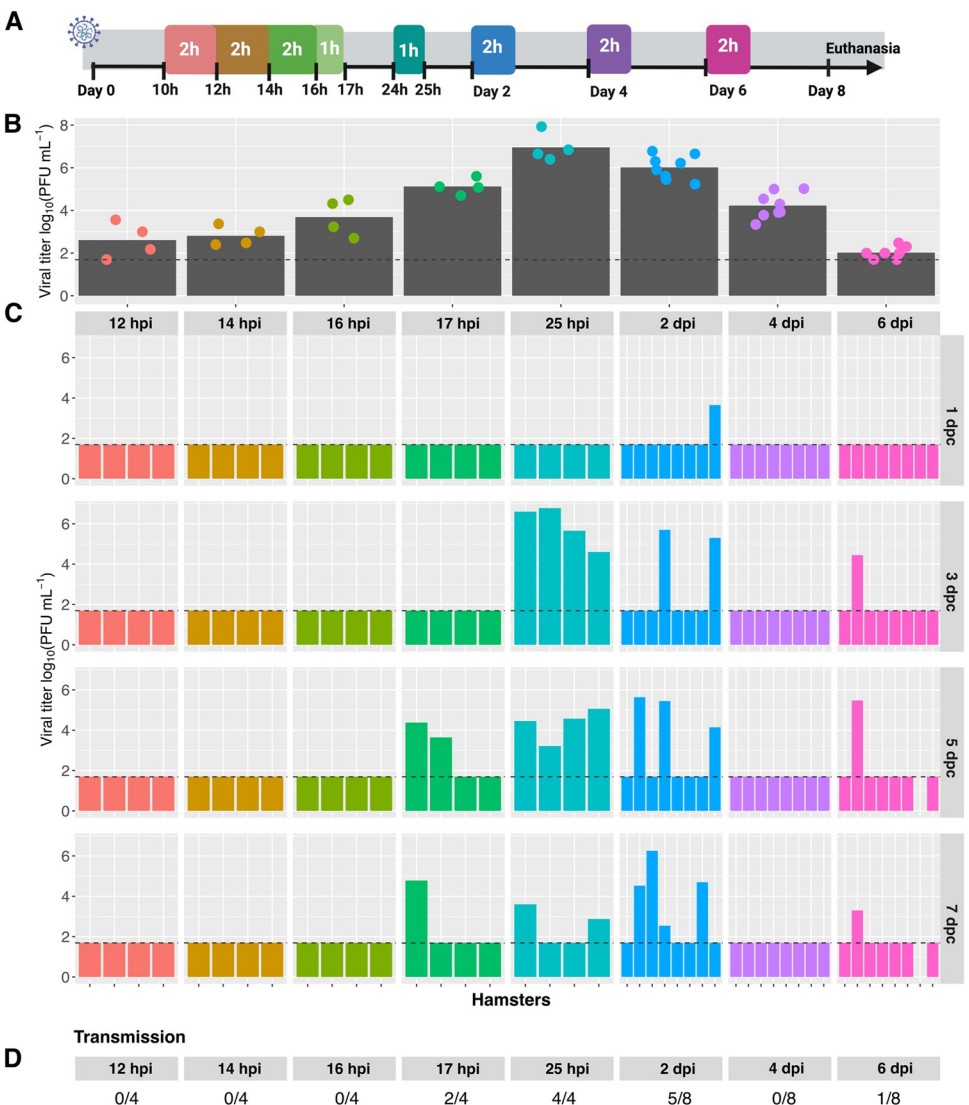

**Fig 2. Period of transmissibility corresponds to period of high viral load in donor animals.** A] Schematic depicting experimental timeline. Animals were inoculated with $1\times10^2$ PFU [titered on VeroE6 cells] at Day 0 and exposures initiated at Day 1. Colored boxes show the different exposure times, with the duration indicated within each box and the start times indicated with black ticks. B] Viral load in inoculated animals at the conclusion of the exposure period. Time points are indicated under the x-axis and a different color is assigned to each different exposure time. Bars show mean viral titer and dots show individual hamsters. C] Viral titers in nasal washes collected from contacts. Different exposure periods are shown with different colors. The time post-inoculation at which the exposure period concluded is shown at the top of each facet. Time points at which nasal wash samples were collected are shown at the right of each row in units of days post contact [dpc]. For the early exposure groups, n = 4 transmission pairs. For the 2 dpi, 4 dpi and 6 dpi exposure groups, data from two independent experiments are displayed together giving total n = 8 transmission pairs. All exposures were carried out at 20˚C and 50% RH. Horizontal dashed line indicates limit of detection [50 PFU]. Missing data indicate that the animal died or was euthanized mid-way through the experiment. D] The fraction of transmission pairs in which recipients shed infectious virus at one or more time points is indicated. Group comparison using Fisher's exact test revealed that differences in transmission were statistically significant [P = 0.0017].

would be detectable for SARS-CoV-2 and therefore evaluated transmission with exposures carried out under dry [20 or 30% RH], intermediate [50% RH] or humid [80% RH] conditions. In each case, temperature was held constant at 20˚C. A schematic of the transmission

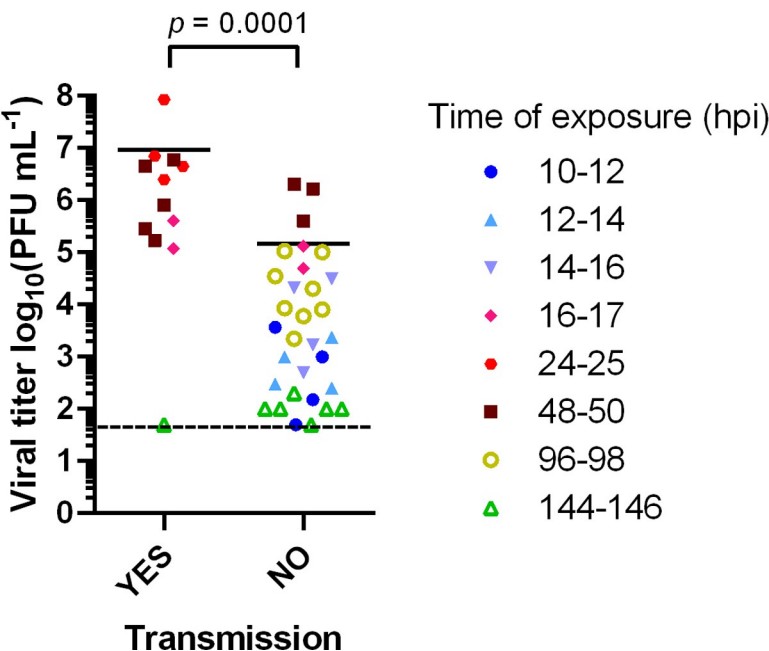

**Fig 3. Transmission is associated with higher donor viral loads.** Data analyzed are also shown in Fig 2. Viral titers detected in nasal lavage samples of donor animals collected at the time of exposure are plotted according to whether transmission occurred or not. All animals were inoculated with $1x10^2$ PFU [titered on VeroE6 cells] and exposures were carried out at 20°C and 50% RH. The time of exposure in hours post-inoculation [hpi] is shown in different colors. Unpaired Student's t-test was performed on log-transformed data.

experimental setup is shown in **Fig 4A**. When a high inoculation dose and a five-day exposure duration beginning at 24 hpi were used, transmission was found to be highly efficient irrespective of humidity [**Fig 4B**]. Reasoning that any effects of humidity may be difficult to detect when overall transmission efficiencies are saturated, we performed similar experiments with a reduced inoculation dose and exposures carried out early after infection when intermediate transmission frequency was observed. Namely, naïve animals were exposed for three hours beginning at 14 h after inoculation of donors. Since placement of the hamsters disrupts the relative humidity in their environment, this timing allowed equilibration of relative humidity during a period when no transmission was observed [**14–16 hpi; Fig 2C**]. A schematic of the transmission experimental setup is shown in **Fig 4C**. Here, results revealed a trend of more transmission under high RH conditions [**Fig 4D**]. These results clearly indicated that, contrary to expectation, low ambient humidity was not favorable for transmission. However, hamsters were subjected to humidity set points for only a brief period, during exposure. If the effects of humidity on transmission act at the level of the host, then they would be unlikely to be manifested in this experimental system. To address this possibility, we tested whether pre-conditioning of animals under different humidity conditions for 4 d prior to inoculation or exposure impacted transmission. For a given group of hamsters, pre-conditioning and exposure were carried out under the same dry, intermediate or humid conditions. A schematic of the transmission experimental setup is shown in **Fig 4E**. Again, the most transmission was seen at high RH [**Fig 4F**]. As before, body weight loss was typically moderate to severe in inoculated animals and more variable for contact animals [**S5 Fig**]. These results suggest that high ambient RH may support SARS-CoV-2 transmission in this model and indicate that low RH does not promote transmission in this system.

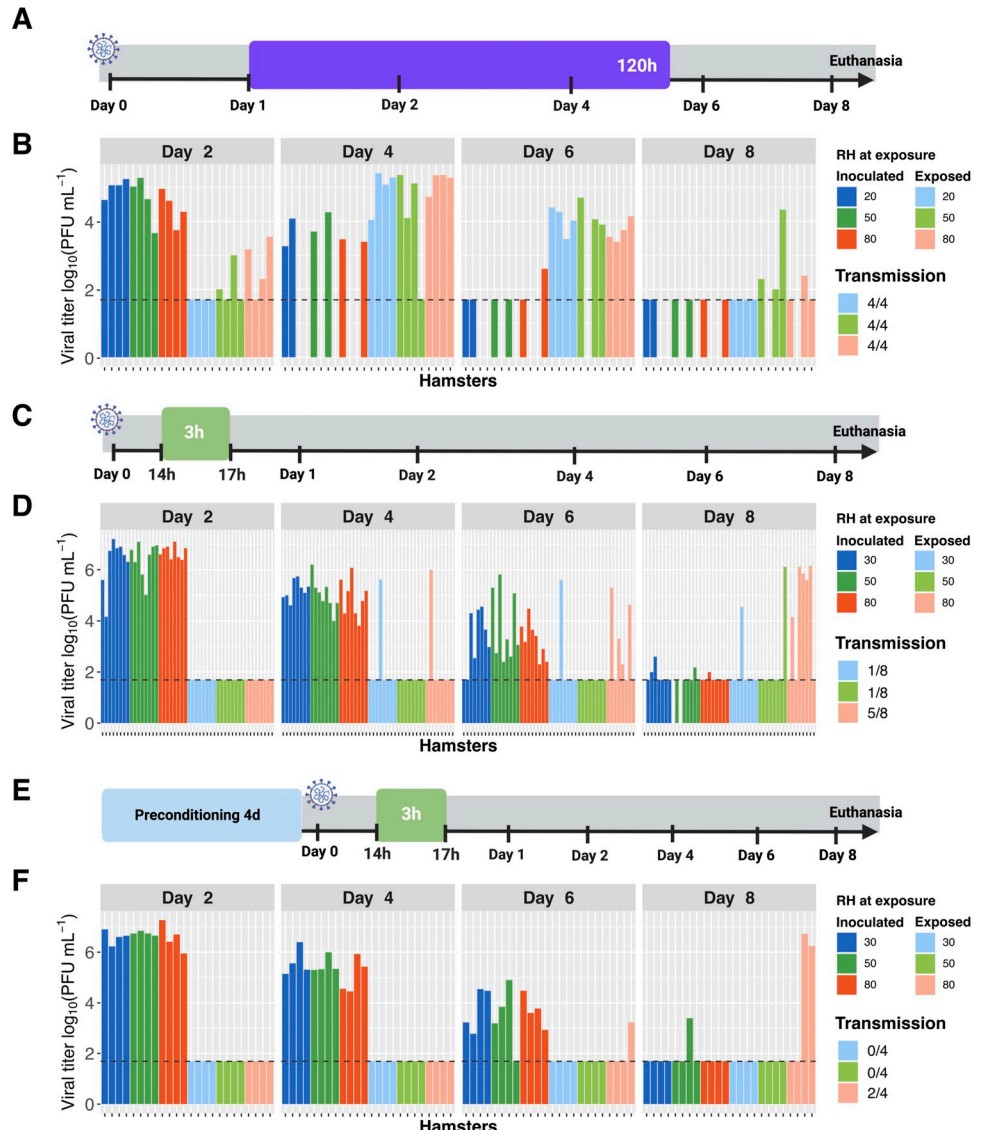

**Fig 4. Low humidity does not promote SARS-CoV-2 transmission in hamsters.** A, C, E] Schematic diagrams depicting experimental timeline. Animals were inoculated at Day 0 and exposures initiated at Day 1. Colored boxes show different exposure conditions, with the duration indicated within each box and the start times indicated with black ticks. B, D, F] Viral titers in nasal lavage samples collected from inoculated [dark colors] and exposed [light colors] hamsters are plotted. Facets show results from 2-, 4-, 6- and 8-days post-inoculation [dpi] with different colors indicating the RH of exposure. A, B] Donor animals were inoculated with 1x10⁴ PFU [titered on VeroE6 cells] and contacts were exposed for a period of five days under the indicated RH conditions, beginning at 24 hpi. N = 4 transmission pairs. C, D] Donor animals were inoculated with 1x10² PFU [titered on VeroE6 cells] and contacts were exposed for a period of 3 h under the indicated RH conditions, beginning at 14 hpi. Data are combined from two independent experiments, giving a total of n = 8 transmission pairs. Comparison between 30% RH or 50% RH and 80% RH using Fisher's exact test revealed that differences were not significant [P = 0.12]. E, F] Donor and contact hamsters were preconditioned to the tested environmental RH for a period of four days. Donor animals were then inoculated with 1x10² PFU [titered on VeroE6 cells] and contacts were exposed for a period 3 h under the indicated RH conditions, beginning at 14 hpi. N = 4 transmission pairs. Comparison between 30% RH or 50% RH and 80% RH using Fisher's exact test revealed that differences were not significant [P = 0.43]. Horizontal dashed line indicates limit of detection [50 PFU]. Missing data indicate that the animal died or was euthanized mid-way through the experiment. Temperature was 20˚C in all experiments shown.

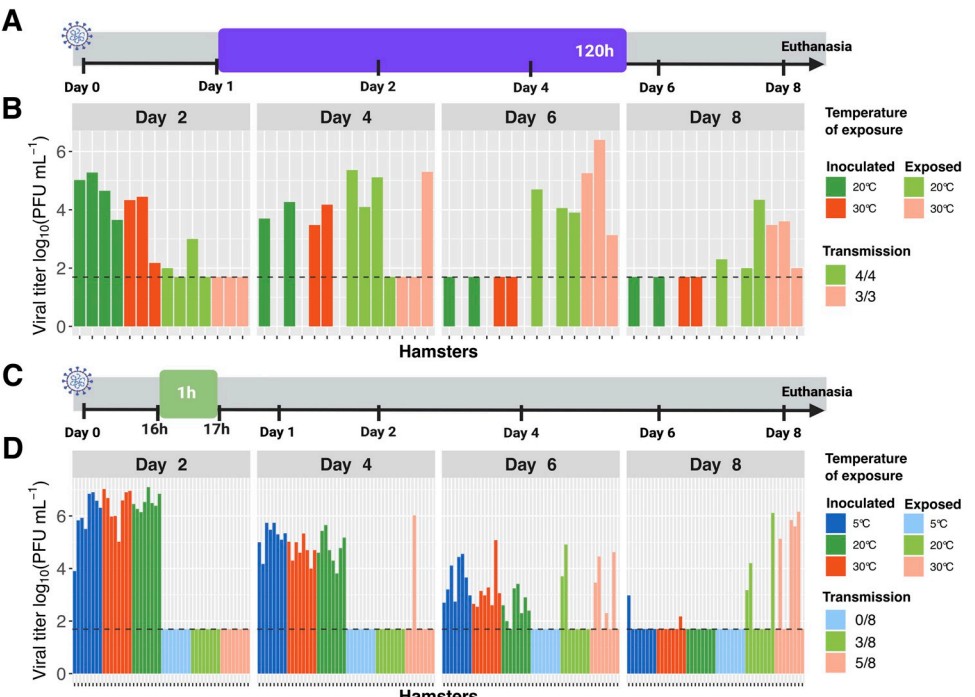

**Fig 5. Low temperature does not promote SARS-CoV-2 transmission in hamsters.** A, C] Schematic depicting experimental timeline. Animals were inoculated at Day 0 and exposures initiated at Day 1. Colored boxes show different exposure times, with the duration indicated within each box and the start times indicated with black ticks. B, D] Viral titers in nasal lavage samples collected from inoculated [dark colors] and exposed [light colors] hamsters are plotted. Facets show results from 2-, 4-, 6- and 8-days post-inoculation [dpi] with different colors indicating the temperature of exposure. A, B] Donor hamsters were inoculated with $1x10^4$ PFU [titered on VeroE6 cells] and contacts were exposed for a period of five days under the indicated temperature conditions, beginning at 24 hpi. N = 4 transmission pairs. Comparison between 20˚C and 30˚C using Fisher's exact test revealed that differences were not significant [P = 1.0]. C, D] Donor animals were inoculated with $1x10^2$ PFU [titered on VeroE6 cells] and contacts were exposed for a period of one hour under the indicated temperature conditions, from 16–17 hpi. Data are combined from two independent experiments, giving a total of n = 8 transmission pairs. Comparison between 5˚C and 20˚C using Fisher's exact test revealed that differences were not significant [P = 0.20]. Comparison between 20˚C and 30˚C using Fisher's exact test revealed that differences were not significant [P = 0.62]. Comparison between 5˚C and 30˚C using Fisher's exact test revealed that differences were statistically significant [P = 0.026]. Horizontal dashed line indicates limit of detection [50 PFU]. Missing data indicate that the animal died or was euthanized mid-way through the experiment. RH was 50% for all experiments shown.

## Minimal impact of temperature on SARS-CoV-2 transmission

As for RH, ambient temperature was previously found to modulate the efficiency of aerosol transmission of influenza A virus, with cold conditions favoring spread [27–29]. We hypothesized that a similar effect would be detectable for SARS-CoV-2 and therefore evaluated transmission with exposures carried out under cold [5˚C], intermediate [20˚C] or warm [30˚C] conditions. In each case, RH was held constant at 50%. A schematic of the transmission experimental setup is shown in **Fig 5A**. When a five-day exposure duration beginning at 24 hpi was used to compare 20˚C and 30˚C environments, transmission was found to be highly efficient under both conditions [**Fig 5B**]. Since our hypothesis was that cold conditions would be favorable, we did not test 5˚C in this system. Instead, we performed similar experiments with a lower inoculation dose and a one-hour exposure beginning at 16 hpi–an approach designed to yield intermediate levels of transmission and thereby allow detection of temperature effects. A schematic of the transmission experimental setup is shown in **Fig 5C**. While intermediate levels of transmission were observed at 20˚C and 30˚C in these experiments, no transmission was

observed at 5°C. Thus, results did not support the hypothesis that cold conditions augment transmission of SARS-CoV-2 [**Fig 5D**]. As seen previously, hamster body weight loss in this series of experiments was moderate to severe in inoculated animals and variable for contact animals [**S6 Fig**].

## Discussion

The COVID-19 pandemic has laid bare the importance of understanding the efficiency and dynamics of respiratory virus transmission. Here we report the results of detailed animal studies designed to quantify the effects of exposure duration, exposure timing and environmental conditions on SARS-CoV-2 transmission. Using a hamster model in which donor and contact animals share air space but are not in direct contact, we find that SARS-CoV-2 transmission is highly efficient, leading to infection of most of recipient animals with exposure periods as short as one hour and under a wide range of humidity and temperature conditions. Peak transmission was observed when exposures were carried out between 16 h and 48 h after inoculation of donor hamsters, revealing an early and narrow window of opportunity for transmission. This period of infectivity corresponded with high viral titers in the nasal tract, implicating viral load as a major driver of transmission.

Substantial evidence has accumulated for pre-symptomatic transmission of SARS-CoV-2 among humans, suggesting that the infectious period begins early after infection [30–33]. However, the timing of infection and any subsequent transmission are often difficult to determine in natural settings. Experimental studies allow these parameters to be determined with precision and our data indicate that onward transmission of SARS-CoV-2 occurs readily after only a brief incubation period of ~16 h. However, in translating this information to humans, it is important to note that the course of viral replication observed in inoculated hamsters was abbreviated relative to that in humans. Clinical data suggest that peak viral loads occur 3–4 days after infection [34,35] and results of an experimental human challenge study similarly show peak loads in the throat at day 4 and in the nose at day 5 post-infection [36]. In our experiments, peak titers were seen in hamsters much earlier, at 24 hpi. This difference of kinetics may relate to initial dose, as these doses of $1x10^2$–$1x10^4$ PFU used here likely exceed a typical natural dose and are higher than that used in the human challenge study. Indeed, examination of viral titers in recipient animals shows delayed kinetics of replication for brief, early exposures–likely associated with lower initial dose–compared to exposures carried out over an extended period or at the peak of donor viral load. As a result, the period of contagiousness identified in hamsters is unlikely to translate directly to human hosts. However, the observation that the period of contagiousness corresponds to times of high viral load is likely to extend to humans. Comparing our data to those from experimentally infected humans [36], we would therefore infer that the period of peak contagiousness in humans is between approximately days 3 and 8 post-infection.

Our data suggest that temporal changes in the potential for transmission likely stem from changes in viral load. While a direct relationship between viral load and transmission potential is intuitive, the extent to which respiratory virus transmission relies on symptoms and is limited by antiviral responses in the donor individual remain active areas of investigation [37–40]. The observation herein that the frequency of transmission declines with viral titers both early and late in the course of infection suggests that viral load is a primary determinant of transmission, irrespective of the host processes that influence viral load. These results are consistent with those of a COVID-19 cohort study which showed a strong relationship between transmission and viral load, independent of symptoms [41]. Viral loads detected in infected individuals vary across several orders of magnitude [34,35]; while timing and method of sample collection likely contribute to this disparity, biological variation appears to be high. The link between

viral loads and transmission is therefore critical to understand. Our data support the notion that heterogeneity in viral load across individuals is a plausible driver of the extreme transmission heterogeneity observed for SARS-CoV-2 [30,42,43].

Similarities between coronaviruses and influenza viruses in their modes of transmission, particle structure and seasonality suggest that similar factors likely shape the transmission of these pathogens [44]. For influenza A virus [IAV], we previously showed that ambient temperature and humidity have a strong impact on transmission in controlled, experimental settings [27–29]. Clear correlations between these meteorological variables and population level influenza activity have also been reported [45,46]. Thus, seasonal changes in humidity and temperature are likely major drivers of influenza dynamics at the population level. These observations led to our hypothesis that SARS-CoV-2 transmission would be suppressed by high ambient humidity and temperature conditions. Our data do not validate this hypothesis and instead reveal efficient transmission among hamsters housed in humid or warm environments. Robust transmission under humid conditions is consistent with the U-shaped relationship between RH and stability that has been reported for multiple enveloped viruses, including SARS-CoV-2 [47–49]. In general, however, these results were unexpected based on reports of super-spreading events in cold or dry indoor environments [50–53] and available information on how temperature and RH modulate viral stability, host susceptibility and aerosol dynamics [49,54–56]. The unexpected results may stem from our approach of performing exposures early in the course of infection; transmission success may be more subject to stochastic effects at early times compared to exposures carried out at times when viral load in donors is at peak levels.

While circulation of endemic coronaviruses shows a clear seasonal pattern [57–59], increased activity in winter has not been a consistent feature of the COVID-19 pandemic. Influenza pandemics also do not follow the typical seasonality of epidemic influenza but have shown a decline in circulation during summer months [57,60], suggesting the relative influence of seasonal drivers is reduced but not ablated for pandemic influenza. In the case of SARS-CoV-2, a lack of seasonal patterns at this early stage in its circulation may reflect low population immunity, dominance of other epidemiological factors such as population behavior and government interventions [15], or a lack of sensitivity to the seasonal drivers that shape the dynamics of epidemic coronaviruses and seasonal influenza viruses. While our data suggest a lack of sensitivity to high humidity and temperature, we caution against over-interpretation of these data given the limitations inherent in our experimental system.

A detailed understanding of the host, viral and environmental factors that shape transmission efficiency is of fundamental importance to efforts to elucidate the drivers of SARS-CoV-2 dynamics across spatial-temporal scales. This knowledge in turn is invaluable for refining strategies to interrupt transmission. Our data reveal that the timing of exposure is a potent determinant of transmission potential and point to viral load as the underlying driver of this effect.

## Methods

### Ethics statement

This study was performed in accordance with the recommendations in the Guide for the Care and Use of Laboratory Animals of the National Institutes of Health. Animal husbandry and experimental procedures were approved by the Emory University Institutional Animal Care and Use Committee (IACUC).

### Virus

SARS-CoV-2/human/USA/GA-83E/2020 was isolated from a clinical sample by culture on VeroE6 cells [60]. The passage 1 virus stock was aliquoted and stored at -80°C. Genomic

analysis verified the predominance of full-length genomes and retention of the furin cleavage site in Spike. Sequencing also revealed the presence of the D614G polymorphism in the Spike protein and six nucleotide deletion in ORF 8 [nucleotides 28,090–28,095; amino acids AGSKS to A—ES]. Viral titers were determined by plaque assay on VeroE6 cells. The passage 1 virus stock was used for all experiments outlined herein.

## Cells

VeroE6 cells were obtained from ATCC [clone E6, ATCC, #CRL-1586] and cultured in DMEM [Gibco] supplemented with 10% fetal bovine serum [Biotechne], MEM nonessential amino acids [Corning] and Normocin [Invivogen]. Cells were routinely confirmed to be negative for mycoplasma contamination. These cells were used for viral culture and titration.

## Animals

All animal experiments were conducted in accordance with the Guide for the Care and Use of Laboratory Animals of the National Institutes of Health. Studies were conducted under animal biosafety level 3 [ABSL-3] containment and approved by the Institutional Animal Care and Use Committee [IACUC] of Emory University [protocol PROTO202000055]. Animals were humanely euthanized following guidelines approved by the American Veterinary Medical Association [AVMA]. Outbred, male, Golden Syrian hamsters of 90–101 g body weight were obtained from Charles River Laboratories and singly housed on paper bedding with access to food and water *ad libitum*. The hamsters were singly housed to avoid stress for the animals since male hamsters kept in pairs tend to fight and can cause injury. Prior to inoculation, nasal lavage and euthanasia, hamsters were sedated with ketamine [120 mg/kg]–xylazine [4 mg/kg] administered intraperitoneally. Xylazine was reversed with 1mg/kg of atipamezole administered intraperitoneally. Animal health was monitored daily through visual observation and determination of body weight in consultation with the clinical veterinarian as dictated by the IACUC protocol.

## Inoculation

Virus was diluted serially in PBS to achieve the desired dose and then sedated hamsters were inoculated intranasally with a 100 μl, applied dropwise to both nares with the animal in dorsal recumbency. Doses ranged from $1x10^2$ PFU to $1x10^4$ PFU [titered on VeroE6 cells], as indicated in figure legends.

## Nasal lavage

Virus was sampled from the upper respiratory tract by nasal lavage. Sedated hamsters were placed in ventral recumbency with nose suspended above an open Petri dish. A total volume of 400 μl PBS was applied to the nares using a micropipette and allowed to drop back into the dish. An additional volume of 200 μl PBS was used to wash the dish. Fluid in the dish was collected, aliquoted and stored at -80˚C prior to determination of viral titers by plaque assay.

## Exposure of naïve hamsters to inoculated hamsters

Exposures were carried out in rodent cages modified through the addition of a double-walled porous barrier, which divided the cage in two. A single hamster was placed on either side of the barrier. The barrier reached from wall-to-wall and floor-to-lid and comprised two stainless-steel sheets placed 14 mm apart, with perforations 4 mm in diameter arrayed across each

sheet. Each side of the cage was supplied with food and water. The cage was enclosed with a filter top.

For all exposures, cages were placed within a Caron 6040 environmental chamber. These chambers allow tight control of humidity and temperature conditions. To achieve uniformity of these conditions throughout the chamber, air flow rates are relatively high, at 13000 L/min. Environmental conditions within the chamber and within the rodent cages [with hamsters present] were verified using a Temperature/Humidity WIFI Data Logger [Traceable Products; Webster, Texas]. Desired conditions were readily met within the hamster cages. After opening the chamber door to place cages, recovery time varied from 10 to 60 minutes, with dry [20 or 30% RH] conditions requiring the longest recovery times. To allow testing of different RH conditions, it was therefore important to place the animals in the chambers at an early time point after inoculation of donors [14 hpi] such that chamber equilibration was achieved prior to the start of the infectious period [16 hpi].

Inoculated donor animals were singly housed within environmental chambers shortly after inoculation. One naïve recipient was introduced on the opposite side of each exposure cage at 14 h– 6 dpi, as indicated in each figure legend. Exposures were carried out for durations ranging from 1 h to 5 d. Where exposures exceeded 24 h, animals were removed from the chambers daily for determination of body weight and, on a subset of days, nasal lavage. To avoid spurious transmission, care was taken during animal handling. Gloves were changed and biosafety cabinet and weighing container were disinfected between animals.

## Data analysis

Data analysis was done using RStudio 1.3.959 and Prism version 6.0.7. Plots were aesthetically modified using Inkscape 1.0. Transmission schematics were created with BioRender.com.

## Supporting information

**S1 Fig. Exposure system.** A] A schematic of cage with inoculated and exposed hamsters on opposite sides of a porous barrier. B] A photo of a representative cage used in this study. Cages were modified with the addition of a porous, double-walled divider. Inoculated and exposed hamsters were placed on opposite sides to evaluate transmission through the air.
(TIF)

**S2 Fig. Body weight loss of inoculated and exposed hamsters corresponding to Fig 1.** Body weights of [A] inoculated [dark colors] and [B] exposed [light colors] are plotted. Animals were inoculated with $1x10^4$ PFU [titered on VeroE6 cells] and exposures initiated at 24 hpi. Exposures were carried out for five days at 30˚C and 50% RH. Missing data indicate that the animal died or was euthanized mid-way through the experiment. Animals were humanely euthanized when weight reached 75% of starting weight.
(TIF)

**S3 Fig. Dynamics of viral load in inoculated hamsters.** Related to Fig 2. Viral titers in nasal lavage samples collected longitudinally from inoculated hamsters are plotted. Each bar represents a hamster. Facets show results from 1-, 2-, 4-, 6- and 8-days post-inoculation with different colors indicating the time at which these donor animals were placed together with contacts. Samples collected on day 1 were collected at the conclusion of the exposure period; thus, the timing of collection varies with the treatment group for the day 1 dataset. All animals were inoculated with $1x10^2$ PFU [titered on VeroE6 cells]. Horizontal dashed line indicates limit of detection [50 PFU]. Missing data indicate that the animal died or was euthanized mid-

way through the experiment.
(TIFF)

**S4 Fig. Body weight loss of inoculated and exposed hamsters corresponding to Figs 2 and 3.** Body weights of inoculated [A, C, E, and G] and exposed [B, D, F, and H] are plotted. Animals were inoculated with $1x10^2$ PFU [titered on VeroE6 cells] and exposures were carried out at 20oC and 50%RH. A different color is assigned to each different exposure period. Missing data indicate that the animal died or was euthanized mid-way through the experiment. Animals were humanely euthanized when weight reached 75% of starting weight.
(TIFF)

**S5 Fig. Body weight loss of inoculated and exposed hamsters corresponding to Fig 4.** Body weights of inoculated [dark colors] [A, C, E, and G] and exposed [light colors] [B, D, F, and H] are plotted. A and B] Animals were inoculated with $1x10^4$ PFU [titered on VeroE6 cells] and contacts were exposed for a period of five days under the indicated RH conditions, beginning at 24 hpi. N = 4 transmission pairs. C-F] Donor animals were inoculated with $1x10^2$ PFU [titered on VeroE6 cells] and contacts were exposed for a period of 3 h under the indicated RH conditions, beginning at 14hpi. N = 8 transmission pairs from two independent experiments. G and H] Donor and contact hamsters were preconditioned to the tested environmental RH for a period of four days. Donor animals were then inoculated with $1x10^2$ PFU [titered on VeroE6 cells] and contacts were exposed for a period of 3h under the indicated RH conditions, beginning at 14 hpi. N = 4 transmission pairs. Missing data indicate that the animal died or was euthanized mid-way through the experiment. Animals were humanely euthanized when weight reached 75% of starting weight.
(TIFF)

**S6 Fig. Body weight loss of inoculated and exposed hamsters corresponding to Fig 5.** Body weight of inoculated [dark colors] [A, C, and E] and exposed [light colors] [B, D, and F] are plotted. A and B] Donor hamsters were inoculated with $1x10^4$ PFU [titered on VeroE6 cells] and contacts were exposed for a period of five days under the indicated temperature conditions, beginning at 24 hpi. N = 4 transmission pairs. C-F] Donor animals were inoculated with $1x10^2$ PFU [titered on VeroE6 cells] and contacts were exposed for a period of one hour under the indicated temperature conditions, from 16–17 hpi. N = 8 transmission pairs from two independent experiments. Missing data indicate that the animal died or was euthanized mid-way through the experiment. Animals were humanely euthanized when weight reached 75% of starting weight.
(TIFF)

## Acknowledgments

We thank Hui Tao and Shamika Danzy for technical assistance and the Emory University Division of Animal Resources for their support with animal care, cage construction and logistical challenges.

## Author Contributions

**Conceptualization:** Chung-Young Lee, Anice C. Lowen.

**Data curation:** Ketaki Ganti, Lucas M. Ferreri, Chung-Young Lee, Camden R. Bair.

**Formal analysis:** Ketaki Ganti, Lucas M. Ferreri, Chung-Young Lee, Camden R. Bair, Anice C. Lowen.

**Funding acquisition:** Anice C. Lowen.

**Investigation:** Ketaki Ganti, Lucas M. Ferreri, Chung-Young Lee, Camden R. Bair.

**Methodology:** Ketaki Ganti, Lucas M. Ferreri, Chung-Young Lee, Camden R. Bair, Gabrielle K. Delima, Kate E. Holmes, Mehul S. Suthar.

**Resources:** Mehul S. Suthar.

**Supervision:** Anice C. Lowen.

**Validation:** Ketaki Ganti, Lucas M. Ferreri, Chung-Young Lee, Camden R. Bair, Gabrielle K. Delima, Kate E. Holmes, Mehul S. Suthar.

**Visualization:** Lucas M. Ferreri, Chung-Young Lee.

**Writing – original draft:** Anice C. Lowen.

**Writing – review & editing:** Ketaki Ganti, Lucas M. Ferreri, Chung-Young Lee, Camden R. Bair, Gabrielle K. Delima, Kate E. Holmes, Mehul S. Suthar, Anice C. Lowen.

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
