## [Decision Letter · Decision Letter 0]

9 Feb 2022

Dear Dr. Lowen,

Thank you very much for submitting your manuscript "Timing of exposure is critical in a highly sensitive model of SARS-CoV-2 transmission" for consideration at PLOS Pathogens. As with all papers reviewed by the journal, your manuscript was reviewed by members of the editorial board and by several independent reviewers. In light of the reviews (below this email), we would like to invite the resubmission of a significantly-revised version that takes into account the reviewers' comments.

We cannot make any decision about publication until we have seen the revised manuscript and your response to the reviewers' comments. Your revised manuscript is also likely to be sent to reviewers for further evaluation.

Sincerely,

Kellie A. Jurado

Associate Editor

PLOS Pathogens

Ana Fernandez-Sesma

Section Editor

PLOS Pathogens

Kasturi Haldar

Editor-in-Chief

PLOS Pathogens

orcid.org/0000-0001-5065-158X

Michael Malim

Editor-in-Chief

PLOS Pathogens

orcid.org/0000-0002-7699-2064

Reviewer's Responses to Questions

**Part I - Summary**

Reviewer #1: In this manuscript, Ganti et al. evaluate how timing of exposure, exposure duration, as well as humidity and temperature impact SARS-CoV-2 transmission in highly controlled experimental settings, using a hamster model of infection.

While the authors found that exposure duration as short as 1h was sufficient to enable viral transmission, the timing of exposure appeared as a critical parameter for successful viral transmission. Especially, authors found that successful viral transmission associated with a time when viral load peaks in the nasal cavity of the animals, that is between 17h to 48h post infection. Finally, they addressed the role of humidity and temperature on viral transmission, and suggest that low temperature and humidity prevent effective transmission of SARS-CoV-2.

This work contributes to fill up an important knowledge gap and bring new lights as to the roles of viral and environmental parameters on SARS-CoV-2 transmission dynamics. Clearly, the value of this work lies within the fact that authors analyze the impact of these different parameters on viral transmission using a well characterized animal model of SARS-CoV-2 infection and highly controlled experimental settings.

However, while this work provides increased clarity as to the relative impact of such parameters on transmission, a fundamental question is whether the reported findings are specific to the model and experimental settings used in this study, or can be directly relatable to human to human transmission.

Reviewer #2: The manuscript by Ganti et al. describes the effects of exposure period, timing, and coincident variables such as relative humidity and ambient temperature on the transmission efficiency of SARS-CoV-2, which was measured by measuring viral load in the nasal washes of Syrian hamsters co-housed (1:1) with infected “donor” hamsters on opposite sides of a porous barrier. The authors first show that exposure periods as short as 1 hour within this system are sufficient for transmission, when the donor hamsters were infected 24 hours prior to exposure of the recipient, or “contact”, hamsters. Next, the authors varied the time post-infection when infected donor hamsters were co-housed with contact hamsters, finding that donor hamsters co-housed in a window of 16 hours – 2 days post-infection were the most likely to transmit. This period also correlated with highest viral load in donor hamsters. Congruously, the authors found that the viral loads of donor hamsters positively correlated with whether contact animals became infected or not. Lastly, the authors varied two environmental variables, relative humidity and temperature, and found that high relative humidity and high temperature each tended to promote SARS-CoV-2 transmission.

Overall, the work credibly evaluates the relationships between transmission of SARS-CoV-2 and exposure timing, viral load of donor individuals, relative humidity, and temperature. Exposure timing corresponding with viral load drives the greatest amount of SARS-CoV-2 transmission. While somewhat expected, this finding confirms that virus load and the timing of peak virus titer is important for transmission. The authors could better link this to peak viral load in humans in the discussion. Similarly, the evaluation of humidity and temperature provide an unexpected result. Both higher humidity and higher temperature were associated with increased transmission, contrasting what was expected based on influenza studies. While the result is clear for both temperature and humidity, it would be beneficial for the authors to speculate a potential mechanism for these differences. Together, the findings shed light on variables affecting SARS-CoV-2 transmission, highlighting the central importance of viral load to transmission.

Reviewer #3: Ganti et al explore the effect of timing of exposure on SARS-CoV-2 transmission in a golden Syrian hamster model of SARS-CoV-2 transmission. As expected, exposure periods as brief as one hour are sufficient to support robust transmission. However, the timing after infection is critical for transmission success, with the highest frequency of transmission to contacts occurring at times of peak viral load in the donor animals. Humidity and temperature had no detectable impact on transmission when exposures were carried out at optimal timing. However, at sub-optimal exposure timing, transmission was improved at relatively higher levels of humidity or temperatures. While high humidity is logical for increased viral particle viability, the high temperature effects on transmission were not expected. The data presented is contradictory to the conclusions drawn in many places of the manuscript, making the manuscript confusing and many of the conclusions incorrect, in my opinion.

**Part II – Major Issues: Key Experiments Required for Acceptance**

Reviewer #1: Despite being rather expected and hardly serving the novelty potential of the study, the major take-away of this work – that the peak of viral load associate with effective transmission – is likely translatable to human transmission and represent as such a comprehensive confirmation of an association between high viral load and high risk of transmission. In contrast, how the subjacent findings related to exposure timing can be harnessed to increase our understanding of human-to-human SARS-CoV-2 transmission remain uncertain (and honestly stated by the authors, line 265) given the specificities of the experimental system and parameters in use in this study.

Most importantly, experiments assessing the impact of different temperature and humidity are relatively confusing. Indeed, as transmission efficiency appears to be dependent of time of exposure and incubation period under similar RH or temperature conditions, the translational nature of these findings and how they may increase our understanding of SARS-CoV-2 transmission between humans is unclear. This is emphasized by evidence that the RH and temperature findings in this study are in contradiction with previous reports, and by the fact that authors admit that their unexpected results may be driven by the specific experimental settings they are using.

Of note, it is unclear to me why the authors used a 1e4 PFU viral inoculum for assessing the impact of high/low temperature and RH in optimal transmission settings (Figure 4A and 5A; 5 days exposure, 24hpi) while using a 1e2 PFU viral dose in the context of the sub-optimal transmission settings (Figure 4B,C and 5B) although they mention in the main text that the experiments between optimal and suboptimal environmental settings were otherwise conducted similarly (“similar” line 177 and 218). It remains unclear to me why authors decided not to use the same viral dose across all experiments, i.e. the one they used in earlier experiments (1e2 PFU) when setting optimal transmission settings. One therefore cannot exclude that a viral dose of 1e4 in Figure 4A and 5A (a dose that is not used when assessing timing of exposure and exposure duration) could have prevented to observe any environmental impact on viral transmission, because it would have been too high of a dose.

Overall, beyond the uncertainty regarding the translational nature of this study’ findings to human transmission, the experimental parameters defining viral transmission under specific environmental settings with the hamster model itself also remain unclear.

Reviewer #2: No additional experiments required

Reviewer #3: Major Concerns:

1. Regarding Figure 1, I completely disagree with the statement “Thus, within the range tested, the period of exposure had minimal impact on SARS-CoV-2 transmission in hamsters.” According to Figure 1, the animals clearly had more transmission events and resulted in higher viral titers when exposed with animals at days 4 or 6 post-inoculation as compared to animals exposed at days 2 or 8 post-inoculation. This needs to be quantified in several ways, including # transmission events, average viral titers of recipient animals, and perhaps a combination of both.

2. Donor animals loose their viral titer significantly after 2 days post-inoculation (dpi). In fact, according to supplementary Figure 2, the inoculated animals have their highest viral loads at 2 dpi. By 4 or 6 dpi, their viral titers are quite lower, and by day 8, the titers are basically zero. This seems to in complete contradiction with the donor animals having their best transmission times at 4 or 5 dpi, according to Figure 1. This is neither described or discussed, which is a major flaw of the manuscript.

3. There is clearly a discrepancy in the results from Fig. 2B-C and Fig. 1, when it comes to the timing of exposure post-donor inoculation. One possibility is that the viral loads are being determined in different ways for the two figures. The most accurate method should be plaque assays. Where viral loads determined via plaque assays for both experiments? If not they should be. Also, these discrepancies need to be explained experimentally and in the Results/Discussion sections. They are very obvious and largely ignored.

4. In Figure 3, both 16-17 and 48-50 hpi are shown as red dots. Times of exposure 0-12 hrs and 12-14 hours are also the same colors, so it is impossible to distinguish between them. Therefore, it is impossible to interpret this figure.

5. I completely disagree with this conclusion regarding the humidity studies in Fig. 4: “These results clearly indicated that, contrary to expectation, low ambient humidity was not favorable for transmission.” I think the field out there agrees fairly well than lower humidity does not favor transmission, while high humidity does.

6. I am not convinced on the temperature results in Figure 5 either, given that the animals were at these temperatures (it seems), which will affect their behavior, movement, etc… and these factors were not controlled.

7. The system used may not as natural as allowing the animals to naturally interact. It does have the value of having a more control exposure experimental system, but it is more artificial as far as how animals would naturally interact. This should at least be discussed in the manuscript.

8. No information for the size of the pores of the walls between chambers is included in the manuscript. This is very important to calculate whether all types of aerosol particles can travel through the walls.

**Part III – Minor Issues: Editorial and Data Presentation Modifications**

Reviewer #1: - As authors established that 5°C exposure prevents transmission under sub-optimal transmission settings, authors should test this temperature under optimal condition to strengthen evidence for its potential impact on transmission.

- Can the authors provide any clinical information from the donor and exposed animals? Were (all) the transmission events inducing weight loss?

- Line 134: Should this write “12, 14 or 16h”?

- Line 135-136: Should this write “beginning at 17h”?

Reviewer #2: 1. A schematic of the transmission experiments (number of animals co-housed) would be beneficial in the main text or in the supplementary text. At a minimum, please state the number of animals (1:1) in the figure legend.

2. In figure 1, while the legends are reasonable, it would be helpful to label the times on the X axis for clarity. There is a significant amount of data/information and cross referencing to the legend is a bit tedious if labels can be added.

3. Supplemental figure 2 could be included in main text. Supplemental figure 1 could be removed.

4. Ideally, a time point at 36 hours would have been helpful. However, the results are clear that peak viral titer (25hr-2DPI) associate with the most robust transmission period. The authors should comment on if a minimum amount of virus is necessary for transmission. Specifically observing differences in transmission at 17HPI and 4 days though titers are similar.

5. The colors in figure 3 need to be modified to be more distinct, especially 16-17 vs 48-50 as the colors cannot be discerned as constructed. With that said, the trend lines are convincing.

6. Figure 4 and 5 could provide additional labels or would benefit from a schematic. It is difficult to decipher exactly what is going on in the figure from the legend. This can be included in the main text or supplement.

7. In the sentence beginning on Line 124, it was unclear whether the timepoints given (e.g. in Line 124) refer to the time when contact animals were co-housed with donor animals or the time when donor animals were sampled for their own viral loads. The sentence beginning Line 127 should be similarly edited for clarity.

8. Figure 3 lacks a line delineating the limit of detection.

9. The phrase in Line 272: "transmission efficiency declines with infectious viral titers" is worded awkwardly and seems to contradict the data.

10. There is a lack of statistical testing in much of the text. A test to compare proportions of infected animals could perhaps add additional rigor.

Reviewer #3: (No Response)

PLOS authors have the option to publish the peer review history of their article (what does this mean?). If published, this will include your full peer review and any attached files.

Reviewer #1: No

Reviewer #2: No

Reviewer #3: No
---

## [Editor Report · Decision Letter 1]

9 Mar 2022

Dear Dr. Lowen,

We are pleased to inform you that your manuscript 'Timing of exposure is critical in a highly sensitive model of SARS-CoV-2 transmission' has been provisionally accepted for publication in PLOS Pathogens.

Best regards,

Kellie A. Jurado

Associate Editor

PLOS Pathogens

Ana Fernandez-Sesma

Section Editor

PLOS Pathogens

Kasturi Haldar

Editor-in-Chief

PLOS Pathogens

orcid.org/0000-0001-5065-158X

Michael Malim

Editor-in-Chief

PLOS Pathogens

orcid.org/0000-0002-7699-2064
---

## [Editor Report · Acceptance letter]

22 Mar 2022

Dear Dr. Lowen,

We are delighted to inform you that your manuscript, "Timing of exposure is critical in a highly sensitive model of SARS-CoV-2 transmission," has been formally accepted for publication in PLOS Pathogens.

Best regards,

Kasturi Haldar

Editor-in-Chief

PLOS Pathogens

orcid.org/0000-0001-5065-158X

Michael Malim

Editor-in-Chief

PLOS Pathogens

orcid.org/0000-0002-7699-2064